# Formation Mechanisms of Rural Summer Health Destination Loyalty: Exploration and Comparison of Low- and High-Aged Elderly Leisure Vacation Tourists

**DOI:** 10.3390/bs12100367

**Published:** 2022-09-28

**Authors:** Puwei Zhang, Shuaifeng Guo, Li Zeng, Xiaoyun Li

**Affiliations:** 1College of City Construction, Jiangxi Normal University, Nanchang 330022, China; 2College of History, Culture and Tourism, Jiangxi Normal University, Nanchang 330022, China

**Keywords:** rural summer health leisure vacation, destination loyalty, elderly leisure vacation tourist, aged group, ABC theory, destination competitiveness

## Abstract

Destination loyalty is a key indicator of the competitiveness of tourist destinations. Rural summer health leisure vacations for urban elderly (RSHLVUE) tourists span a wide range of ages. Destination operators need to understand the loyalty formation mechanisms of different aged tourists. RSHLVUE tourists were divided into a low-aged group (LA) and a high-aged group (HA) to examine and modify the hypothesis of the relationship between perceived value, tourist well-being, place attachment, and destination loyalty based on affect, behavior, and cognition (ABC) theory. The test results of the measurement model indicate that the HA showed stronger responses in terms of cost value, sense of meaning, and place dependency. The formation mechanism of destination loyalty for the LA is tourist well-being → perceived value → place attachment → destination loyalty, and for the HA is perceived value → tourist well-being → place attachment → destination loyalty. The findings deepen the understanding of destination loyalty among elderly leisure vacation tourists and can guide RSHLVUE destination managers to enhance destination competitiveness.

## 1. Introduction

The increasingly aging population and the implementation of the rural revitalization strategy have produced a new leisure vacation industry in China, namely, rural summer health leisure vacations for the urban elderly (RSHLVUE). By the end of 2021, China had 267.36 million people aged 60 and above, accounting for 18.9% of the total population [1]. A comfortable summer temperature for the elderly is around 26–28 degrees Celsius [2] However, most cities on the southeast side of the Heihe–Tengchong Line, which account for more than 90% of China’s total population [3], have a high number of uncomfortable summer days [4]. The retired elderly in these cities have a good economic base and considerable free time [5], so they have the demand for summer vacation and leisure. High-altitude mountainous areas are naturally endowed with comfortable temperatures, high vegetation cover, and fresh natural air for developing RSHLVUE, but these mountainous areas are often inaccessible and have no pillar industries to promote their economic development. With the implementation of the rural revitalization strategy, the transportation infrastructure in these mountainous areas has been improved and the conditions for developing RSHLVUE are available.

RSHLVUE has the characteristics of rural, summer, health, and elderly tourism while developing its own unique characteristics as follows: (1) Tourists are mainly elderly people from nearby cities, (2) the stay length of tourists is usually 1–3 months, (3) the purpose of the vacation is for summer health, (4) the operators of destinations are mainly local villagers, and (5) the destination includes a combination of food and accommodation with a low consumption price. RSHLVUE has been the leading industry in some villages in China.

Loyal tourists informally connect potential tourists such as friends and colleagues to the destination [6]. The cost of retaining existing tourist groups is much less than that of developing new ones [7]. Studies have explored destination loyalty for the following tourism types: international tourism [8], film-induced tourism [9], sports tourism [10], culinary tourism [11], heritage tourism [12], and sustainable tourism [13]. However, the mechanisms of destination loyalty for RSHLVUE tourists have not been specifically explored.

Different age groups of tourists perceive the external environment differently [14], and the factors and mechanisms that influence destination loyalty may also differ. The World Health Organization classifies middle-aged and older people by age: the middle-aged group (45–59 years), young–old group (60–74 years), old–old group (75–89 years), and longevous group (>90 years) [15]. However, existing loyalty studies mostly consider tourists as a whole, without dividing them further. According to our pre-survey, the age range of RSHLVUE tourists is large. Thus, destination operators need to understand the loyalty formation mechanism of different aged tourists to provide targeted services for different aged groups of tourists.

Affect, behavior, and cognition (ABC) is a theory of attitudes in consumer behavior [16]. It argues that the order of occurrence among affect, behavior, and cognition may change the roles they play. One of the most widely used models is the standard learning hierarchy model (CAB framework), such as online shopping loyalty [17], restaurant loyalty [18], and tourist loyalty [19]. On the basis of the CAB framework, this study proposes the RSHLVUE destination loyalty formation mechanism by considering perceived value as the cognitive dimension, tourist well-being and place attachment as the affective dimension, and destination loyalty as the behavioral dimension. The mechanisms of loyalty formation for different aged tourists can provide theoretical guidance for the subsequent development of RSHLVUE.

Based on the characteristics of RSHLVUE tourists, this study has two objectives. First, how to measure the four variables of perceived value, tourist well-being, place attachment, and destination loyalty. Second, to construct the hypothesis of influence mechanism for these four variables and to validate/revise the hypothesis of influence mechanism with the questionnaire data from low- and high-aged elderly leisure vacation tourists from Zhongyuan Township, Jing’an County, Jiangxi Province, China.

## 2. Literature Review and Hypothesis Development

### 2.1. RSHLVUE

RSHLVUE represents an overlap between rural, summer, health, and elderly tourism. RSHLVUE destinations are usually in high-altitude mountain villages with the characteristics of rural tourism [20]. RSHLVUE tourists usually stay in a destination close to their place of residence for 1–3 months to spend the hot summer months, which falls under the category of summer tourism [21]. The main objective of RSHLVUE tourists is also for their own health, which carries the characteristics of health tourism [22]. RSHLVUE’s main source of tourists is urban retired elderly, thus characterizing elderly tourism [23].

Some of the villages that are developing RSHLVUE have already formed a certain scale. For example, Guzhu Village in Shuikou Town, Changxing County, Huzhou City, Zhejiang Province receives 2.01 million elderly tourists from cities such as Shanghai and Hangzhou every year [24]; Sanping Village in Zhongyuan Township, Jing’an County, Yichun City, Jiangxi Province receives 600,000 summer tourists mainly from Nanchang City every year [25]; and Huangshui Town in Shizhu County, Chongqing City receives more than 200,000 daily summer tourists mainly from Chongqing City during the peak summer season [26].

### 2.2. Perceived Value and Tourist Well-Being

Similar to Dodds et al. [27], the perceived value of RSHLVUE tourists can be thought of as a subjective evaluation of the services [28], social bonding [29], facilities [30], and environmental experiences [31] of the rural leisure vacation destination compared with the costs incurred. Tourist well-being is the sum of positive emotions and feelings of value that tourism activities bring to a tourist [32]. Dimensions of tourist well-being include satisfaction [33], positive emotions [34], sense of meaning [35], and sense of achievement [36]. When elderly tourists feel better about their health, their positive emotions are stronger. Thus, the health experience is an expression of the well-being of RSHLVUE elderly tourists (see Appendix A for details).

When tourists’ perceived value of a destination is high, their level of well-being is high as well [37]. Tourists’ perceived value has been confirmed to affect their well-being. For example, Zhang et al. concluded that perceived value is an important antecedent of customers’ subjective well-being [38]. Liu et al. found that the emotional, functional, and social values of perceived value directly influence well-being [39]. Junaid et al. argued that the quality, emotional, and knowledge values of tourists’ perceived value influence well-being through brand love [40]. Li et al. confirmed that tourists’ perceived value positively influenced their subjective well-being [41]. Therefore, the first hypothesis of this study is:

**H1.** *Perceived value has a significant positive effect on tourist well-being*.

### 2.3. Perceived Value and Place Attachment

The concept of place attachment emphasizes the emotional connection between people and places [42]. Scholars consider that place attachment encompasses emotional and social components [43]. In the RSHLVUE context, place attachment refers to the special emotional relationship that elderly leisure vacation tourists have with destination villages based on the physical environment and social relationships [44].

Positive and strong attachments are created by staying in a place for a long period [45]. Lee et al. argued that tourists’ perceptions of positive experiences have a positive effect on tourists’ post-tour evaluations and help foster place attachment [46]. Wang et al. found that perceived value has a significant positive effect on place attachment [47]. Zhang et al. showed that perceived value is an important antecedent of place attachment and activity attachment [48]. On this basis, the following hypothesis is proposed:

**H2.** *Perceived value has a significant positive effect on place attachment*.

### 2.4. Perceived Value and Destination Loyalty

Loyalty is a firm commitment to repurchasing a product or service in the future [48]. Destination loyalty is usually assessed from attitudinal and behavioral perspectives [49]. Loyalty is measured in terms of a tourist’s intention to revisit or recommend the destination to others [50]. Tourist perception is an important antecedent of destination loyalty [51,52] and has a significant positive effect on tourists’ intention to revisit [53]. Liu et al. found that emotional, functional, and social values in perceived value indirectly influence tourist loyalty through the mediating effect of well-being [39]. Lee et al. argued that tourists’ perceived values of service and social well-being contribute to positive post-trip loyalty [46]. Thus, we propose that tourists’ perceived value is related to destination loyalty as follows:

**H3.** *Perceived value has a significant positive effect on destination loyalty*.

### 2.5. Tourist Well-Being and Place Attachment

RSHLVUE tourists usually stay in the countryside destination for a month or more. The countryside’s good natural environment, comfortable climate, and fresh air enhance the physical health of elderly leisure vacation tourists. The group living and the richness of activities could also eliminate their loneliness and enhance their well-being [54]. They form an emotional attachment to the destination, that is, place attachment [55]. Bogdan et al. revealed a significant positive relationship between place attachment and tourist well-being [56]. Vada et al. indicated that tourist well-being acts as a moderating variable for memorable tourism experiences and place attachment [57]. Basu et al. suggested that the relationship between place attachment and well-being is direct and significant [58]. Trinanda et al. indicated that destination attachment can be enhanced by memorable travel experiences and hedonic well-being [59]. To this end, the fourth proposition of this research is developed as follows:

**H4.** *Tourist well-being has a significant positive effect on place attachment*.

### 2.6. Tourist Well-Being and Destination Loyalty

Tourist well-being is the most powerful determinant of positive behavioral intentions. For example, Jamaludin et al. found that relatively stable and happy groups were important in ensuring destination loyalty intentions [60]. Vada et al. concluded that hedonic well-being has a significant impact on revisit intentions and positive word-of-mouth [61]. Kim and Kim argued that psychological well-being significantly influences loyalty [62]. Reitsamer and Brunner–Sperdin revealed that tourist well-being has a significant positive effect on their intention to revisit and their willingness to engage in positive word-of-mouth about a destination [37]. Liu et al. asserted that tourist well-being acts as a mediator between perceived value and destination loyalty, directly influencing destination loyalty [39]. The fifth hypothesis of this study is as follows:

**H5.** *Tourist well-being has a significant positive effect on destination loyalty*.

### 2.7. Place Attachment and Destination Loyalty

Place attachment is seen as an important push for tourists to destinations and is an important factor influencing tourists’ behavioral intentions [63]. The higher the level of place attachment to a tourist destination, the more loyal to it they tend to be [64]. Horakova et al. suggested that consumers who have a strong attachment to a place spread positive word-of-mouth about it regardless of the environment [65]. Gautam concluded that vacation satisfaction, place attachment, and emotional experience are significant predictors of destination loyalty [66]. Han et al. indicated that place attachment indirectly influences revisit intention through attitude, subjective norms, and perceived behavioral control [67]. On the basis of the previous literature, we posit that place attachment contributes to destination loyalty as follows:

**H6.** *Place attachment has a significant positive effect on destination loyalty*.

### 2.8. Preliminary General Conceptual Framework

The CAB framework in ABC theory considers the consumer decision-making process for a product to be similar to a problem-solving process, with the following steps [68,69]. (1) Consumers develop their own perceptions of goods or services through their own accumulated experience. (2) They form their own feelings (emotions) through the evaluation of these perceptions. (3) They behave accordingly to their emotions. The cognitive dimension in this study is perceived value, the affective dimension is tourist well-being and place attachment, and the behavioral dimension is destination loyalty, and the initial conceptual model *M*_0_ of RSHLVUE destination loyalty is proposed, as shown in Figure 1.

## 3. Methodology

### 3.1. Research Sites

The research site for this study was Zhongyuan Township, Jiangxi Province, China. It is about 100 km from Nanchang, the capital of Jiangxi Province. Its average altitude is 655 m, air humidity is 82.5%, forest coverage is more than 86%, and air has a negative oxygen ion content of 100,000 ions per cubic centimeter. Its average summer temperature is 20–22 °C, which is 6–10 °C lower than that of Nanchang City. Zhongyuan Township has been developing RSHLVUE for more than 20 years. By 2020, the destination had more than 640 nongjiales (Farmhouse Joy restaurants), with more than 20,000 beds and 2000 employees. The comprehensive income from summer vacation tourism amounted to 130 million yuan. The main tourists come from Nanchang. Thus, Zhongyuan Township provides a convenient venue for researchers to observe, extract, and analyze the mechanisms of destination loyalty formation among elderly tourists.

### 3.2. Survey Procedure

First, to identify the measurement dimensions of perceived value, tourist well-being, place attachment, and destination loyalty, a systematic literature review and the first semi-structured interviews were conducted. The first semi-structured interview was conducted from 1–4 July 2021 in Zhongyuan Township with 15 elderly tourists. Each participant was interviewed for about 45 min.

Second, a questionnaire survey was conducted. The survey was carried out by 14 students trained in research methods. Owing to eyesight problems, most elderly people were interviewed verbally for the survey. The survey was conducted in two phases: (1) A pre-survey was conducted from 22–23 July 2021; (2) a formal survey was conducted from 25 July–5 August 2021.

To reduce the effect of common method biases, a questionnaire was designed with three sets of questions in different orders. The Harman one-way test and the error variable control method test were conducted referring to Podsakoff et al. [70] and Meade et al. [71]. In the Harman one-way test, a common method bias exists if a single factor accounting for more than 50% of the variance is present [70]. In the error variable control method, a common method bias exists if the model fit metrics do not become better with the inclusion of a common variance bias latent variable [71].

Finally, the second semi-structured interviews were conducted from 25–31 August 2021. Twenty tourists were interviewed, ten of whom were in the 45–74 age group and ten in the >75 age group. The average interview time for each interviewee was 45 min.

### 3.3. Measures

In this study, the measures of the four main constructs—perceived value, tourist well-being, place attachment, and destination loyalty—were drawn from previous research. The items for the five dimensions of perceived value were adapted from the existing literature [72,73,74]. The items of two dimensions related to place attachment were extracted from the existing literature [75,76]. The two items for destination loyalty were drawn from previous studies [77]. The items for emotional experience and sense of meaning in tourist well-being were adapted from previous studies [78,79], and the items for health experience in tourist well-being were derived from research summaries (See Appendix A).

First, to refine the 52 preliminary items measuring the four constructs from literature and summaries, the expert panel method was applied. The expert panel was composed of three professors of health and wellness tourism research. In this study, the Delphi expert consultation was conducted using back-to-back expert anonymous questionnaires, and a combination of online and offline methods was used to collect data. The investigators explained to each invited expert to ensure that they understood the content of the items and the scoring principles. The experts were asked to rate the importance of each initial dimension on a five-level Likert scale. The initial dimension list with the mean of importance assignment > 3.5, the coefficient of variation < 0.25, and the full score ratio > 20% were selected to form the second version of the initial dimension group. Experts were asked to classify the initial items into the second version of the initial dimension list, and they found that 80% of the items were categorized consistently. Thus, the final initial dimension was determined. In the end, nine items were removed.

Second, based on the modification suggestions collected during the pre-survey phase, 16 items were deleted, and 8 items were modified for clarity and comprehensibility. All items were scored using a five-point Likert scale ranging from strongly disagree (1) to strongly agree (5) (See Appendix B). The final measurement items are shown in Table 1.

## 4. Analysis and Results

### 4.1. Respondents’ Profile and Common Methodological Bias Test Results

A one-to-one pilot survey was conducted with 100 respondents, and 92 valid questionnaires were returned. A total of 700 questionnaires were distributed during the formal survey phase, and 641 valid questionnaires were returned, with a valid return rate of 91.57%. The demographic characteristics of the respondents are shown in Table 2.

A common methodological bias test was conducted on the formal survey data. The Harman one-way test showed 11 factors, with the major factor accounting for 29.1%. The results of the error variable control method test showed that the model fit metrics did not become better with the inclusion of the common variance bias latent variable. The common methodological bias was confirmed to not have a significant effect on the results of this study.

### 4.2. Differences in the Four Constructs of Different Aged Groups

A one-way ANOVA test was conducted to understand the differences in the dimensions of the four constructs among elderly leisure vacation tourists of different aged groups; *p* < 0.05 was considered statistically significant. Table 3 shows that the dimensions of the four constructs differed significantly across the aged groups of the leisure vacation tourists. The results of the post hoc LSD test indicate that high-aged elderly leisure vacation tourists show stronger responses in terms of cost value, sense of meaning, and place dependency.

The stronger responses of high-aged elderly leisure vacation tourists in the value of cost can be explained in two ways. On the one hand, the older leisure vacation tourists are, the higher the probability of becoming sick. High-aged leisure vacation tourists consider more expenses for medical care [80]. As a result, they prefer the cheaper and more cost-effective RSHLVUE. On the other hand, prices rise faster than pensions increase, potentially bringing down their consumption levels [81]. Concerning the sense of meaning, the high-aged elderly are often more respected and humbled by low-aged elderly during their travels due to their rich experiences and can therefore develop a stronger sense of meaning [82]. In terms of place dependence, leisure vacation tourists over 80 years old showed the strongest result. This may be due the high-aged elderly leisure vacation tourists preferring familiar environments and being less adaptable to new ones [83].

Based on the above significant differences between low- and high-elderly people, we divided the elderly tourists into a low-aged group (LA) for 45–74 years and a high-aged group (HA) for 75 years and above according to WHO’s age classification criteria.

The measurement and structural models of the four constructs of the RSHLVUE destination loyalty formation mechanism were analyzed separately for the low- and high-aged groups.

### 4.3. Measurement Model Validation and Correction

To validate the measurement models for the four constructs in this study, construct validity, convergent validity, and discriminant validity were assessed using samples from the entire aged group (EA) (*n* = 641), the LA (*n* = 329), and the HA (*n* = 312). As shown in Table 4, the Cronbach’s α for all three sample groups was greater than 0.7 for each construct and its dimensions, and the composite reliability (CR) values were greater than 0.8. This result indicates that the measurement model has good reliability and internal consistency. Kaiser–Meyer–Olkin indices were all greater than 0.7 and Bartlett’s spherical test was significant at the 0.001 level, indicating good overall construct validity. The factor loadings for each dimension’s items were all greater than 0.5, with the majority exceeding 0.7, and the average variance extracted (AVE) values were all greater than 0.5, indicating high convergent validity. The correlation coefficient between any two dimensions was less than the square root of each dimension’s own AVE (see Appendix C), so the measurement model had good discriminant validity. Thus, the reliability and validity of the constructs and their dimensions in the measurement model were satisfied.

### 4.4. Structural Model Validation and Correction

#### 4.4.1. Low-Aged Group

The *M*_0_ was validated using the data from the LA sample, and paths H1, H3, H4, and H5 were not significant. This result indicates that the effect of tourist well-being on destination loyalty is not significant. However, a strong correlation was found between tourist well-being and destination loyalty [39]. Additional semi-structured interviews were conducted to investigate the reasons for the result from 26–31 August 2021.

From the interviews, we know that the LA are more inclined to be involved in activities and travel around than the HA. The better their physical condition is, the stronger their ability to perceive the various elements of the places they visit. Because of the improved health experience in the tourist well-being construct, LA had a better perceived value. The LA are in the stage of social role transition, and their sense of emptiness is more intense than that of the HA [84]. A significant proportion of the LA relieve the emptiness by traveling. This means that LA have more anticipation and generally higher positive emotions in the pre-tourism period compared with the HA. This positive emotion facilitates the participation of the LA in various activities during the leisure vacation, in turn affecting the perceived value of the destination. Sreejesh and Ponnam suggested that emotions can influence perceived value [85]. Kiviniemi et al. argued that emotions and perceived value are in an interactive relationship, not a one-way influence [86]. Therefore, this study proposes the following additional hypothesis.

**H1′.** *Tourist well-being has a significant positive effect on perceived value*.

On the basis of *M*_0_, hypothesis H1 was replaced with H1′ to form *M*_1_. *M*_1_ was examined by the LA sample data, and the results showed that H1′ was established. To improve the fit index of the model, *M*_1_ was modified to obtain *M*_2_ as follows. (1) The cost value dimension was removed, as the perceived value to cost value significance level (*p*-value) was greater than 0.001 and had a factor loading of 0.18 (<0.3). (2) The sense of meaning dimension was removed because the *p*-value of the tourist well-being to a sense of meaning was greater than 0.001 and a factor loading of 0.29 (<0.3). (3) The association of EME2 with FUV3 and PLD1 with SEV2 was established because these two groups had the highest residual correction indices. The fit indices of models *M*_1_ and *M*_2_ were compared (Table 5), and *M*_2_ could fit the sample data better. Figure 2 shows the influence relationship between the constructs of *M*_2_ and their influence path.

The results showed that the *p*-values of H1′, H2, and H6 were significant, and the three hypotheses were accepted. Additionally, the *p*-values of H3, H4, and H5 were not significant, so the three hypotheses were rejected (Table 6).

In terms of total variance explained values, 97% of perceived value can be explained by tourist well-being ($R^{2}$ = 0.97), 86% of place attachment can be explained by perceived value ($R^{2}$ = 0.86), and place attachment explains 32% of destination loyalty ($R^{2}$ = 0.32). The mechanism of LA leisure vacation destinations loyalty is that perceived value acts as a mediator of tourist well-being and place attachment, which then indirectly influences destination loyalty through place attachment.

#### 4.4.2. High-Aged Group

The $M_{0}$ was validated with data from the HA sample, and the model fit results were good. To further optimize the model, the association of EME2 with FUV3 and PLD1 with SEV2 was established to obtain $M_{3}$. The fitted index of $M_{3}$ is better than that of $M_{0}$ (Table 7). Figure 3 reveals the influence relationship between the constructs and the influence path. This study also tests H1′, whose model fit index results are not satisfactory. Therefore, the H1 is more suitable for HA.

The empirical results showed that the *p*-values of H1, H4, and H6 were significant, and the three hypotheses were accepted. The *p*-values for H2, H3, and H5 were not significant, and the three hypotheses were rejected (Table 8). In terms of the total variance explaining the values, perceived value explained 78% of tourist well-being ($R^{2}$ = 0.78), tourist well-being explained 86% of place attachment ($R^{2}$ = 0.86), and 50% of destination loyalty was explained through place attachment ($R^{2}$ = 0.50). Perceived value through the two continuous mediating variables, tourist well-being and place attachment, indirectly affect destination loyalty.

### 4.5. Comparison of the Low- and High-Aged Groups

The results from Section 4.3 and Section 4.4 show a significant difference between the LA and HA. First, the measurement model differences were demonstrated on the basis of the factor loadings of the two sample groups under the same construct as follows. (1) Perceived value construct: The environmental value dimension and the social value dimension were more important for the LA. The effect of the cost value dimension was only significant in the HA. The service value and functional value dimensions were more important in the HA. (2) Tourist well-being construct: The emotional experience dimension had a greater effect on the HA, the effect of the sense of meaning dimension was only significant in the HA, and the effect of the health experience dimension did not differ significantly for the LA and HA. (3) Place attachment construct: The effects of place attachment and place identity dimensions were greater in the HA. (4) Destination loyalty construct: The willingness to revisit is stronger for the LA, while the willingness to recommend is higher for the HA.

Second, in terms of structural models, significant differences were found as follows. (1) LA: tourist well-being → perceived value → place attachment → destination loyalty. (2) HA: perceived value → tourist well-being → place attachment → destination loyalty.

## 5. Conclusions and Discussion

### 5.1. Conclusions

Based on ABC theory, this study is the first to explore the formation mechanism of RSHLVUE destination loyalty by aged grouping of elderly leisure vacation tourists. The results confirm that perceived value, tourist well-being, and place attachment are important antecedents of destination loyalty. However, the role of these antecedents in the formation mechanism of destination loyalty varies across aged groups. The specific findings are as follows. (1) In the RSHLVUE context, significant differences exist in the dimensions of each construct in the destination loyalty formation mechanism of LA and HA, with the most significant differences in the cost value dimension of the perceived value construct and the sense of meaning dimension of the tourist well-being construct. (2) Data from the LA and HA samples confirmed that place attachment directly affects destination loyalty. (3) Data from the HA sample confirmed the effect of tourist well-being and place attachment as continuous mediating variables that transmit perceived value on destination loyalty. (4) Data from a sample of the LA confirmed that tourist well-being indirectly influences place attachment through perceived value.

### 5.2. Discussion

#### 5.2.1. Theoretical Implications

This study makes the following theoretical contributions. First, based on ABC theory, tourist well-being is introduced into the study of the formation mechanism of RSHLVUE destination loyalty, enriching the study of destination loyalty formation mechanism. Existing studies usually consider satisfaction an antecedent of destination loyalty, neglecting the affective factor of tourist well-being. The introduction of tourist well-being in this study allows for a better exploration of the factors influencing the psychological dimension of destination loyalty. In addition to combining the characteristics of RSHLVUE tourists, this study constructs three dimensions of well-being measurement: emotional experience, sense of meaning, and health experience, to provide a scale basis for subsequent research related to the well-being of RSHLVUE tourists.

Second, this study further subdivides elderly leisure vacation tourists by age to explore the loyalty formation mechanism, deepening the study of elderly leisure vacation destination loyalty. Existing studies explored the mechanisms of destination loyalty formation mainly from the perspective of tourists in general, without an in-depth understanding of tourist heterogeneity. This study confirms significant differences in the mechanisms of destination loyalty formation between LA and HA. The findings deepen the understanding of destination loyalty among RSHLVUE tourists and enrich the knowledge related to gerontology.

#### 5.2.2. Management Implications

The local government of the destination can formulate relevant policies to guide the operators to provide services for LA or HA, which will be more beneficial to improve the satisfaction and loyalty of tourists. More importantly, the results and findings of the study provide the following management insights and guidance to destination operators. First, leisure vacation tourists’ place attachment to the destination should be fostered. The results showed that destination loyalty of LA and HA was directly influenced by place attachment. Their functional needs for leisure vacation destinations can be met by improving medical support facilities and providing humanized care services to enhance their local dependence. Their sense of identification with leisure vacation destinations can be enhanced by conducting more recreational activities.

Second, more targeted initiatives can be implemented to enhance the RSHLVUE tourist perceived value. For the LA, more focus should be placed on the environmental and social interaction aspects, such as by designing themed accommodations and adding small ornamental plants to enhance their perception of environmental value. By organizing group activities, a platform for interactive friendships among LA is provided to enhance their perception of social value. For HA, the focus on service quality, functional facilities, and price should be increased, such as by providing more cost-effective packages or group purchase discounts to improve the perception of cost value. The provision of facilities such as free medical testing equipment, fitness facilities, and walking aids enhances the perception of functional value. Regular free medical checkups should be held and services such as free errands for medicine should be provided to enhance the perception of service value.

Third, the following differentiated strategies can be used to enhance the well-being of LA and HA. For LA, enhancing their well-being at the initial stage of arrival at the destination is needed. Destination operators focus on enhancing the emotional experience and health experience of the LA. Operators can choose to quickly mobilize the positive emotions and health experience of LA when they first arrive at the destination by organizing small-scale activities or giving them health care gifts. For HA, enhancing the sense of well-being in the middle and later stages of arriving at the destination is necessary, and more attention should be paid to the formation of their sense of meaning. Operators should pay more attention to giving respect to HA in their services and give them regular encouragement to make them feel their own sense of meaning in leisure vacations and promote the formation of loyalty.

Finally, different measures can be used to enhance the attractiveness and competitiveness of the destination for the LA and HA. The recommendation willingness of HA is stronger, and RSHLVUE destination operators can be more biased to enhance their word-of-mouth marketing. Operators can give away brochures to them when they leave and ask them to promote the destination. The willingness to revisit is stronger among LA. Destination operators can establish a membership system to enhance their willingness to revisit through membership points and other membership tracking services, such as inviting members to participate in local festivals and events.

#### 5.2.3. Limitations and Suggestions for Future Research

This study has some limitations, which can be addressed by future research. First, the data source of this study was only from Zhongyuan Township, China, potentially hindering the generalizability of the findings. Leisure vacation tourists from different geographical and cultural backgrounds may understand certain topics differently, and future research should involve different regions as well as different countries. Second, this study only uses age as a segmentation criterion and does not examine the variability of destination loyalty under other segmentation criteria. Factors such as marital status and health status may be considered subdivision criteria in subsequent studies. Finally, subsequent studies could attempt to introduce additional variables into the model to further explore more potential antecedents affecting RSHLVUE leisure vacation destinations’ loyalty.

## Figures and Tables

**Figure 1 behavsci-12-00367-f001:**
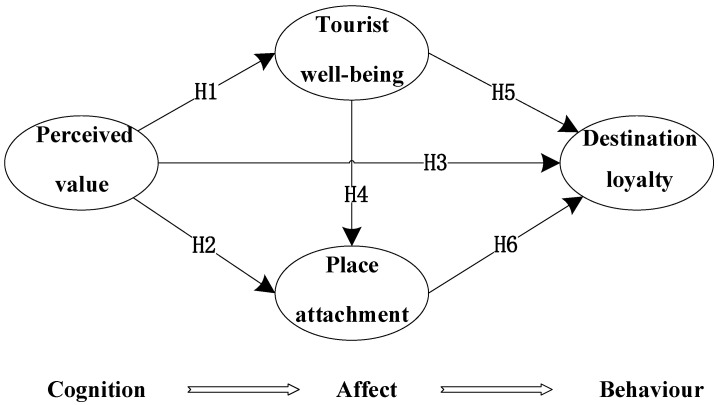
Initial conceptual model *M*_0_.

**Figure 2 behavsci-12-00367-f002:**
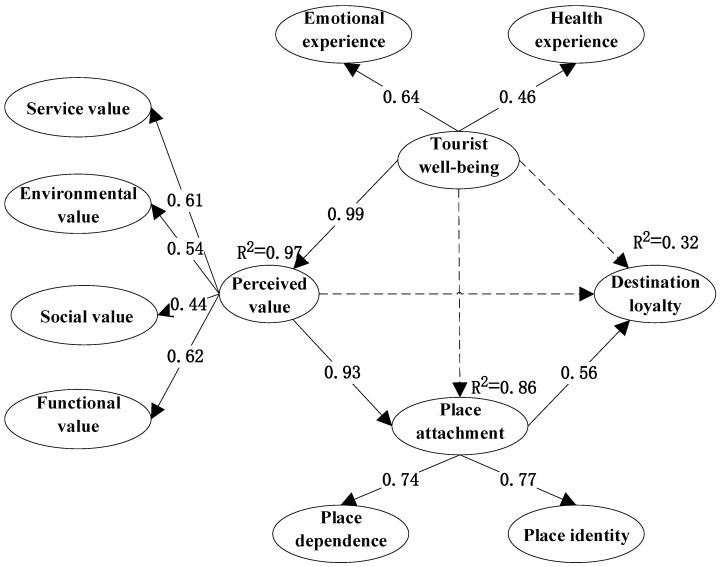
The estimated structural model *M*_2_ for the LA.

**Figure 3 behavsci-12-00367-f003:**
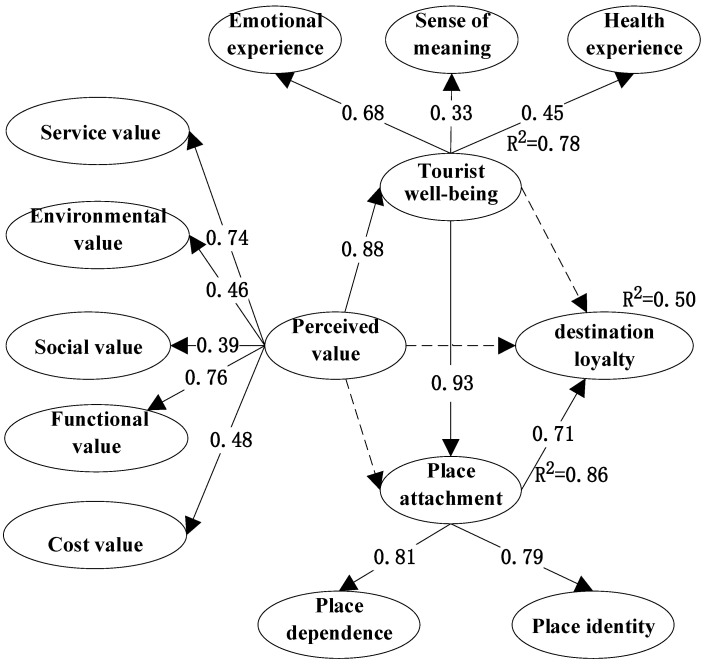
Estimated structural model *M*_3_ for the HA.

**Table 1 behavsci-12-00367-t001:** Measurement items for each construct.

Constructs	Dimension	Items
**PEV** Perceived value	**SEV** Service value	**SEV1** A warm and friendly service staff
		**SEV2** Tourist needs can be addressed in a timely manner
		**SEV3** Paying attention to the special needs of individual tourists and providing assistance
	**ENV** Environmental value	**ENV1** The countryside is well ecologically protected
		**ENV2** The climate is comfortable and pleasant
		**ENV3** The atmosphere is harmonious
	**SOV** Social value	**SOV1** The travel has enhanced the bonding with fellow tourists
		**SOV2** The travel gained the respect and recognition from others
		**SOV3** New friendships made through travel
		**SOV4** A travel experience that you don’t normally feel in life
	**COV** Cost value	**COV1** Reasonable prices for catering
		**COV2** Fair and reasonable accommodation prices
		**COV3** Its worth the cost and effort
	**FUV** Functional value	**FUV1** Well-equipped infrastructure and service facilities
		**FUV2** Better accommodation available
		**FUV3** Good complementary catering available
**TOW** Tourist well-being	**EME** Emotional experience	**EME1** Stunned by the beauty of the scenery during the travel
		**EME2** The pleasure of tasting good food
		**EME3** Excited by the experience of different local cultures
		**EME4** Everyone had fun and no worries during the travel
	**SEM** Sense of meaning	**SEM1** Learned new things and improved abilities
		**SEM2** Increased experience during the travel and the enrichment of life experiences
		**SEM3** Mental state changes during the travel
		**SEM4** Experience a sense of accomplishment during the travel
	**HEE** Health experience	**HEE1** The environment here makes me feel better
		**HEE2** This place makes me mentally free of sickness
		**HEE3** The environment here makes me sleep better
**PLA** Place attachment	**PLD** Place dependence	**PLD1** Compared to other places, this place meets my needs better
		**PLD2** This place gave me a travel experience that cannot be replaced by other places
		**PLD3** This is my favorite place for summer health tourism
		**PLD4** There is more satisfaction to be gained from travel than anywhere else
	**PLI** Place identity	**PLI1** I have strong identification with this place
		**PLI2** This place is of great importance to me
		**PLI3** It is a place where I can be better at being true to myself
**DEL** Destination loyalty		**DEL1** I will revisit for the next trip
		**DEL2** I will recommend my friends and relatives to travel here

**Table 2 behavsci-12-00367-t002:** Sample characteristics.

Variable	Frequency	Percentage	Variable	Frequency	Percentage	Variable	Frequency	Percentage
**Gender**			4–6	140	21.9%	3000–4999	386	60.2%
Male	273	42.6%	7–9	69	10.7%	5000–6999	109	17.0%
Female	368	57.4%	>9	10	3.2%	7000–8999	24	3.7%
**Age**			**Education**			≥9000	8	1.2%
41–50	3	0.5%	Primary school and below	123	19.2%	**Number of days of residence**		
51–60	56	8.7%	Junior high school	171	26.7%	≤15	34	5.3%
61–70	174	27.1%	High school/Technical secondary school	167	26.1%	15–30	118	18.4%
71–80	254	39.6%	College/Undergraduate	172	26.8%	30–60	451	70.4%
>80	154	24.0%	Masters degree and above	8	1.2%	60–90	36	5.6%
**Marital status**			**Number of children**			≥90	2	0.3%
Unmarried	2	0.3%	0	2	0.3%	**Health status**		
Married	517	80.7%	1	184	28.7%	Very bad	3	0.5%
Divorced	2	0.3%	2	232	36.2%	Not good	30	4.7%
Widowed	120	18.7%	3	132	20.6%	Generally good	169	26.4%
**Times of revisit** **to the same village**			≥4	91	14.2%	Good	340	53.0%
0	146	22.8%	**Monthly income (yuan, CNY)**			Very good	99	15.5%
1–3	266	41.5%	≤3000	114	17.8%			

**Table 3 behavsci-12-00367-t003:** Summary of the variability analysis for dimensions in the four constructs.

Demographic Variables	Category	Dimensions	One-Way ANOVA Test		LSD Post Hoc Test
			F-Value	*p*-Value	
**Age**	①41–50 ②51–60 ③61–70 ④71–80 ⑤>80	**Service value**	3.017	0.520	-
		Environmental value	1.861	0.116	-
		Social value	0.411	0.800	-
		Cost value	0.809	0.018	⑤ > ④③②①
		Functional value	1.174	0.321	-
		Emotional experience	1.255	0.286	-
		Sense of meaning	4.710	0.001	⑤ > ④③②①
		Health experience	0.890	0.469	-
		Place dependence	2.486	0.042	⑤ > ④③②①
		Place identity	0.176	0.951	-
		Destination loyalty	0.251	0.909	-

**Table 4 behavsci-12-00367-t004:** Results of the measurement model analysis.

Items	Factor Loading			Cronbach’s α			CR			AVE		
	EA	LA	HA	EA	LA	HA	EA	LA	HA	EA	LA	HA
**PEV**	**-**	**-**	**-**	**0.858**	**0.840**	**0.866**						
SEV1	0.968	0.969	0.966	0.946	0.942	0.949	0.948	0.944	0.950	0.860	0.851	0.866
SEV2	0.948	0.943	0.953									
SEV3	0.862	0.852	0.870									
ENV1	0.937	0.931	0.940	0.852	0.841	0.864	0.863	0.853	0.873	0.680	0.662	0.698
ENV2	0.731	0.727	0.735									
ENV3	0.793	0.769	0.819									
SOV1	0.728	0.714	0.745	0.824	0.814	0.834	0.842	0.834	0.851	0.575	0.561	0.592
SOV2	0.754	0.732	0.782									
SOV3	0.894	0.887	0.896									
SOV4	0.635	0.641	0.631									
COV1	0.931	0.934	0.927	0.945	0.949	0.942	0.948	0.951	0.942	0.860	0.867	0.851
COV2	0.878	0.878	0.880									
COV3	0.969	0.978	0.958									
FUV1	0.597	0.558	0.597	0.818	0.849	0.873	0.843	0.828	0.843	0.648	0.625	0.648
FUV 2	0.856	0.844	0.856									
FUV 3	0.925	0.923	0.925									
**TOW**	**-**	**-**	**-**	**0.831**	**0.828**	**0.842**						
EME1	0.944	0.954	0.935	0.972	0.976	0.972	0.973	0.975	0.971	0.899	0.905	0.893
EME2	0.916	0.927	0.904									
EME3	0.940	0.937	0.943									
EME4	0.991	0.987	0.995									
SEM1	0.959	0.964	0.950	0.895	0.900	0.887	0.899	0.904	0.891	0.694	0.706	0.675
SEM2	0.830	0.831	0.826									
SEM3	0.666	0.674	0.653									
SEM4	0.851	0.865	0.830									
HEE1	0.922	0.922	0.923	0.846	0.847	0.845	0.857	0.851	0.856	0.670	0.659	0.670
HEE2	0.681	0.681	0.656									
HEE3	0.835	0.814	0.852									
**PLA**	**-**	**-**	**-**	**0.857**	**0.845**	**0.868**						
PLD1	0.678	0.635	0.721	0.864	0.852	0.875	0.866	0.858	0.880	0.621	0.605	0.648
PLD2	0.807	0.780	0.829									
PLD3	0.876	0.895	0.857									
PLD4	0.778	0.780	0.806									
PLI1	0.860	0.874	0.851	0.793	0.777	0.809	0.810	0.800	0.823	0.590	0.575	0.609
PLI2	0.658	0.636	0.681									
PLI3	0.773	0.745	0.800									
**DEL**	**-**	**-**	**-**									
DEL1	0.869	0.869	0.853	0.800	0.788	0.809	0.804	0.804	0.810	0.673	0.673	0.681
DEL2	0.769	0.769	0.796									

**Table 5 behavsci-12-00367-t005:** Fitting indicators of model *M*_1_ and *M*_2_.

Fitting Index	Absolute Fit Indices				Relative Fit Indices				Parsimony Fit Indices	
	$\chi^{2}/df$	GFI	RMASE	SRMR	AGFI	NFI	CFI	IFI	AIC	CAIC
Ideal index	1–3	≥0.8	<0.1	<0.08	≥0.8	≥0.9	≥0.9	≥0.9	the smaller the better	
$\mathrm{Model}\;M_{1}$	3.091	0.785	0.080	0.077	0.753	0.825	0.874	0.875	1962.593	2384.647
$\mathrm{Model}\;M_{2}$	1.99	0.868	0.055	0.067	0.842	0.944	0.949	0.950	866.211	1201.935

**Table 6 behavsci-12-00367-t006:** Validation analysis results of model *M*_2_.

Hypothesis	Impact Path	Standardized Path	S.E.	*p*	Result
H1′	Tourist well-being → Perceived value	0.987	0.000	***	Established
H2	Perceived value → Place attachment	0.930	0.093	***	Established
H3	Perceived value → Destination loyalty	−1.442	5.345	0.648	No
H4	Tourist well-being → Place attachment	1.191	2.007	0.632	No
H5	Tourist well-being → Destination loyalty	0.978	4.517	0.721	No
H6	Place attachment → Destination loyalty	0.564	0.169	***	Established

Note: S.E. = standard error; *** *p* < 0.001.

**Table 7 behavsci-12-00367-t007:** Comparison of fit indices for model *M*_0_ and *M*_3_.

Fitting Index	Absolute Fit Indices				Relative Fit Indices				Parsimony Fit Indices	
	$\chi^{2}/df$	GFI	RMASE	SRMR	AGFI	NFI	CFI	IFI	AIC	CAIC
Ideal index	1–3	≥0.8	<0.1	<0.08	≥0.8	≥0.9	≥0.9	≥0.9	the smaller the better	
$\mathrm{Model}\;M_{0}$	2.763	0.790	0.075	0.083	0.758	0.839	0.890	0.891	1773.253	2190.637
$\mathrm{Model}\;M_{3}$	2.070	0.829	0.059	0.074	0.803	0.926	0.932	0.933	1372.695	1785.336

**Table 8 behavsci-12-00367-t008:** Validation analysis results of model *M*_3_.

Hypothesis	Impact Path	Standardized Path	S.E.	*p*	Result
H1	Perceived value → Tourist well-being	0.875	0.085	***	Established
H2	Perceived value → Place attachment	0.402	0.181	0.059	No
H3	Perceived value → Destination loyalty	0.305	0.329	0.180	No
H4	Tourist well-being → Place attachment	0.925	0.108	***	Established
H5	Tourist well-being → Destination loyalty	0.149	0.475	0.613	No
H6	Place attachment → Destination loyalty	0.714	0.137	***	Established

Note: S.E. = standard error; *** *p* < 0.001.

## Data Availability

The data analyzed in this paper are proprietary, and therefore cannot be posted online.

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
