# Peer review of "Formation Mechanisms of Rural Summer Health Destination Loyalty: Exploration and Comparison of Low- and High-Aged Elderly Leisure Vacation Tourists"

_behavsci, 2022, doi:10.3390/bs12100367_

Round 1
Reviewer 1 Report
Overall this paper is well written; Please recheck some of the points.
Abstract: Need to improve, Paper is well written, but the abstract part needs to improve.
Introduction: At the end of the background section, please write the clear objectives of the study, even the authors explain the gap related to Rural summer health leisure vacation for the urban elderly (RSHLVUE).
LR. Please provide statistics related to Urban tourists and how important this sector is for research.
The theory part also needs to explain in detail; before the framework, please write the theory part separately.
The research design well explain.
Discussion, Please recommend some of the policies related to tourism
Author Response
We sincerely thank the reviewer 1 for his/her expert review of the manuscript. Based on his/her comments, we have made further revisions to the previous version of the manuscript. Our point-by-point responses to his/her comments or suggestions are listed below.
Comment 1: Overall this paper is well written; Please recheck some of the points.
Authors' response: Thank you for the comment. We are grateful for your confirmation of our manuscript and, more importantly, for your specific suggestions for its further improvement. We have given careful thought to all your specific suggestions and have carefully revised the manuscript. The specific revisions are described below.
Comment 2: Abstract: Need to improve, Paper is well written, but the abstract part needs to improve.
Authors' response: Thank you for your revision tips. To further clarify the background of the study and the importance of the study, we have added the following sentence to the abstract: “Destination loyalty is a key indicator of the competitiveness of tourist destinations.”, “ Destination operators need understand the loyalty formation mechanism of different aged tourists.”
Comment 3: Introduction: At the end of the background section, please write the clear objectives of the study, even the authors explain the gap related to Rural summer health leisure vacation for the urban elderly (RSHLVUE).
Authors' response: In response to your suggestion, we have added the following research objectives at the end of the introduction: “Based on the characteristics of RSHLVUE tourists, this study has two objectives. First, how to measure the four variables of perceived value, tourist well-being, place attachment, and destination loyalty. Second, to construct the hypothesis of influence mechanism for these four variables and to validate/revise the hypothesis of influence mechanism with the questionnaire data from low- and high-aged elderly leisure vacation tourists from Zhongyuan Township, Jing’an County, Jiangxi Province, China.” The objective of this study has become more prominent.
Comment 4: LR. Please provide statistics related to Urban tourists and how important this sector is for research.
Authors' response: In section 2.1, we add an overview paragraph on the statistics of urban elderly tourists: “Some of the villages that are developing RSHLVUE have already formed a certain scale. For example, Guzhu Village in Shuikou Town, Changxing County, Huzhou City, Zhejiang Province, receives 2.01 million elderly tourists from cities such as Shanghai and Hangzhou every year [24]; Sanping Village in Zhongyuan Township, Jing'an County, Yichun City, Jiangxi Province, receives 600,000 summer tourists mainly from Nanchang City every year [25]; Huangshui Town in Shizhu County, Chongqing City, receives more than 200,000 daily summer tourists mainly from Chongqing City during the peak summer season [26].” The relevant statistics cited further illustrate the importance of rural summer health studies of elderly urban tourists.
Comment 5: The theory part also needs to explain in detail; before the framework, please write the theory part separately.
Authors' response: We have explained the initial conceptual framework as follows. “The CAB framework in ABC theory considers the consumer decision-making process for a product to be similar to a problem-solving process, with the following steps [68,69]. (1) Consumers develop their own perceptions of goods or services through their own accumulated experience. (2) They form their own feelings (emotions) through the evaluation of these perceptions. (3) They behave accordingly to their emotions. The cognitive dimension in this study is perceived value, the affective dimension is tourist well-being and place attachment, and the behavioral dimension is destination loyalty, and the initial conceptual model of RSHLVUE destination loyalty is proposed, as shown in Fig.1.” The preliminary theoretical framework is explained additionally and is separated from Section 2.7 as Section “2.8 Preliminary general conceptual framework”. We corrected an error in the previous version of the manuscript that grouped the overall model framework in section 2.7.
Comment 6: The research design well explain.
Authors' response: Thank you again for your approval of the research design.
Comment 7: Discussion, Please recommend some of the policies related to tourism
Authors' response: Relevant policy recommendations for the government have been added to the discussion section. However, since the research object of this study is tourists, the implications of the study's findings for management practices apply primarily to destination operators, so few policy initiatives can be proposed to governments.
References
- Yuan, F.; Sheng, L.; Lixiong, Z.; Yichen, R. Research on rural endowment villages from multi-dimensional niche perspective: A case study of Northwest Zhejiang Province. Economic Geography 2019, 39, 160-167. http://doi.org/https://doi:10.15957/j.cnki.jjdl.2019.08.019 (in Chinese)
- Zhu, Z.K.. The green wellness, Jing'an Life. Xiaoxiang Morning Post 2020. Available online: https://baijiahao.baidu.com/s?id=1686228623007074711&wfr=spider&for=pc (accessed on 23 September 2022) (in Chinese)
- Lu, X.M.; Zheng, H.. Wellness Stone Pillar: from the "itch" of yellow water to the "nourishment" of cold water, Chongqing Economic Times 2018. Available online: https://e.chinacqsb.com/html/201808/13/node_004.html (accessed on 23 September 2022) (in Chinese)
- Bricker, K. S.; Kerstetter, D. L. An interpretation of special place meanings whitewater recreationists attach to the South Fork of the American River. Tourism Geographies 2002, 64, 396-425. http://doi.org/10.1080/14616680210158146
- Chih, W.; Hsu, L.; Ortiz, J. The antecedents and consequences of the perceived positive eWOM review credibility. Industrial Management & Data Systems 2020, 120, 1217-1243. http://doi.org/10.1108/IMDS-10-2019-0573

Reviewer 2 Report
The paper focuses on a rather unique but upcoming topic of RSHLVU. Investigating the tourism motivation, flow and products for elderly people is always interesting and useful, especially in the developed countries of the Western world. So, the paper can be regarded as original.
The title is appropriate, covering and reflecting the content. The abstract is compact and comprehensive, the key words are well selected.
The introduction is well written, showing the background and importance and highlighting the research goals and questions.
The literature review is perfect. The hypotheses are defined based on the literature review.
The methodology is well selected and well demonstrated, detailed and clearly.
The analysis is executed appropriately and supported by the methodology and literature. The conclusions are based on the results, the discussion is logical and useful.
LImitations of the research are demonstrated in the last part.
I recommend it for publication without any changes.
Author Response
Thank you very much for your approval of our manuscript. Your approval makes us feel more confident about the subsequent research on this topic. Your approval is also the motivation for us to persist in the subsequent research on this topic. Once again, we sincerely thank you.
Reviewer 3 Report
The main goal of the publication is to analyze the loyalty mechanisms of rural summer health centers, including the exploration and comparison of older and older recreational tourists. The work was prepared very carefully with the use of scientific methods. The theoretical part is a literature review prepared on the basis of many sources. The research part is the presentation of the results, analyzes and conclusions from the research. In part of the methodology, the authors presented the method of carrying out research, analysis, and hypotheses, referring to the literature - theory and other research. Based on the literature and experiences from other studies, the authors created their own approach to the problem. Research material based on research carried out (as described) in accordance with the procedures: pilot, research, data reduction, etc. In the analytical part, the authors use advanced methods, formulating models based on data estimation and trying to interpret them. The concluding part is additionally valuable, in which the authors presented, apart from discussions and conclusions, some theoretical and practical implications. Interesting work, mainly in terms of the method of problem analysis and data analysis. The work meets the requirements of an academic publication.
Author Response

(The authors gave the same response as above.)
